# Impact of Climate Change on Productivity and Technical Efficiency in Canadian Crop Production

Viktoriya Galushko *[ID] and Samuel Gamtessa

Economics Department, University of Regina, Regina, SK S4S 0A2, Canada; samuel.gamtessa@uregina.ca
* Correspondence: viktoriya.galushko@uregina.ca

**Abstract:** There is a wide consensus that throughout the 20th century climate has changed globally, with many parts of the world facing increases in average temperatures as well as an increased frequency and intensity of extreme weather events. While the existing climate models can predict future changes in climate with a high degree of confidence, the potential impacts of climate change on agricultural production and food security are still not well understood. In this work, we investigate the link between climate change, output, and inefficiency in Canadian crop production using provincial data for the period of 1972–2016. This study has built a unique climate dataset from station-level weather data and uses a panel stochastic frontier model to explore the effect of climatic conditions on crop production and inefficiency. The results reveal that climatic variables are significant predictors of both the maximum potential output (frontier) and technical inefficiency. The combined effect of higher temperatures and lower precipitation, as reflected in a lower Oury index, is a downward shift of the crop production frontier. While greater variability of daily temperatures during the growing season is found to have no statistically significant effect in the frontier equation, greater variation in rainfall results in a downward frontier shift. The results also show that weather shocks measured as a deviation from historical weather normals are significant predictors of technical inefficiency.

**Keywords:** climate change; crop production; stochastic frontier; technical inefficiency

## 1. Introduction

Throughout the 20th century, global average temperature increased by 1.1 °F [1]. In many parts of the world, increases in average temperatures have also been accompanied by greater inter-annual and seasonal variations in temperatures and precipitation. Climate variation has already been identified as the major cause of year-to-year fluctuations in agricultural production in both developed and developing countries [2]. It is expected that global warming will occur at a faster pace in the next two to three decades, with many studies projecting an increased frequency and severity of extreme weather events [3–5]. Understanding the impacts of past trends on agricultural production is essential to help predict and possibly mitigate the near-term impacts of climate change.

Unlike other production processes, agricultural production is a biological process that is directly influenced by the amount of rainfall and air temperatures. In some parts of the globe, global warming has extended the length of the growing season and allowed earlier planting. For example, according to the US Environmental Protection Agency, the average length of the growing season in the contiguous 48 states has increased by more than two weeks since the beginning of the 20th century [6]. While some studies have shown that warmer temperatures can benefit plant growth due to an increased number of frost-free days, others found that in warmer regions increased temperatures have led to less-than-optimum growing conditions as well as hastened the maturation of crops, thus resulting in lower yields [7,8]. In addition to direct influence, changes in climatic conditions also affect agricultural production indirectly by changing the conditions for the emergence of diseases and pathogens that undermine crop production [9]. Due to

such a very strong link between climate and agricultural output, climate change impact on agricultural productivity and food systems is a research area gaining scholarly interest.

The literature that examines agricultural productivity is rich. While estimation techniques vary, the common theme in existing studies exploring the determinants of agricultural productivity is that they attempt to measure how much output is produced from given amounts of inputs [10–12]. What is frequently omitted from the estimation, however, is the fact that weather is an essential input into the agricultural production process. The omission of environmental production conditions leads to biased estimates of the parameters describing the production technology [13].

Most of the existing literature linking agricultural productivity and climate change incorporates climate and weather variables into the production function together with other inputs. However, climatic conditions also interact with other inputs as their use is likely to be dependent upon the weather. Producers employ a wide array of adaptation strategies, including the adoption of new technological advancements, improvements in irrigation systems, regional shifts in crop acreage and crop species, adjustments in the amounts of inputs used, and changes in crop and livestock management practices. Some studies also recognize that farmer's behaviour is an important element in how climate and weather affect productivity. The fact that producers adapt to climatic conditions implies that input use can be determined by weather. A number of studies have addressed this problem by estimating production possibility frontier models and incorporating climatic variables as determinants of technical efficiency [13–15], where technical efficiency is determined by the difference between the observed ratio of combined quantities of an entity's output to input and the ratio achieved by best practice.

While there is a consensus that climate change has been occurring globally, the climate change patterns and the extent of those changes have varied across continents and countries. This, combined with the fact that agricultural production practices vary across countries and that the variety of crops produced in different countries are contingent upon different climate types (i.e., tropical, dry, temperate, continental), the impact of climate change on agricultural productivity and inefficiency should be assessed in the context of local climate conditions. Therefore, the focus of this study has been narrowed down to Canada.

To date, to the authors' best knowledge, no study has estimated a stochastic frontier function for the Canadian crop sector that incorporates climate variables. To fill this gap, this paper attempts to measure the effect of the local thermal environment on crop productivity and technical inefficiency. Employing provincial-level data on output and inputs from 1972 to 2016, coupled with various climatic variables constructed from finely scaled climate data, we estimate the relationship between climate/weather and technical inefficiency using a stochastic production frontier model. This approach allows us to separate the impact of production inputs such as labour, capital, land, and intermediary inputs from the impact of climatic variables on crop output and inefficiency.

The results show that weather variables are important determinants in both the frontier and inefficiency equations. More specifically, higher temperatures coupled with lower precipitation and greater variability of precipitation throughout the growing season shift the crop production frontier down. We also find that it is unexpected weather shocks, measured by deviations from historical normals, that set producers further away from the production frontier, while the changes in the mean level of weather variables do not have such a pronounced effect on inefficiency. The variability of both temperature and precipitation during the growing season have the opposite impacts on inefficiency, with the former increasing inefficiency and the latter reducing it.

The remainder of this paper is organized as follows. Section 2 sets out a theoretical framework for the estimation of the relationship between weather variables and productivity and efficiency in crop production. Section 3 reviews the data used to support our empirical analysis. In this section, we emphasize the uniqueness of the constructed dataset that is used to capture climatic crop growing conditions at the level of Canadian provinces. Section 4 presents the findings of our analysis. Here, the discussion is focused largely on

key variables—weather variability, average temperatures, and precipitation. The paper concludes with a summary of our results and a discussion of potential policy implications.

## 2. Materials and Methods

### 2.1. Brief Overview of Canada's Changing Climate

Most of Canada's farmland is located in the Prairie region (Manitoba, Saskatchewan, Alberta), with most crop production taking place predominantly in these three provinces. The kinds of crops grown in different Canadian provinces are determined by local climatic conditions. The Prairie region is characterized by a relatively harsh (cold) climate and as a result is not well suited for the production of fruit and field vegetables. In Saskatchewan, for example, climatic conditions are suitable for the production of wheat, canola, and lentils; in Alberta, the major crops are canola, wheat, and barley; in Manitoba, potato production takes second place. The milder climate in British Columbia allows a large production of fruit and field vegetables, while climatic conditions in eastern Canada are favourable for the production of corn, apples, and blueberries.

Similar to the rest of the world, Canada's climate has been changing, with Canada witnessing increases in both temperature and precipitation, and this will likely affect crop production. It has been reported that since 1948, Canada's annual average temperature has increased by 1.7 °C [16,17]. For this study, we have computed growing season average temperature and total precipitation from daily data provided by the weather stations across different census agricultural regions (CARs). As is discussed in greater detail in Section 2.5 and Appendix A, for temperature, daily data were averaged to arrive at monthly averages, which were then averaged across the months of the growing season (May–August). For precipitation, total precipitation during the growing season was calculated as the sum of the daily values. What we can see from the collected data is that climate change patterns have not been uniform across different CARs and provinces. For example, based on the collected climatic data for Alberta, there is a linear trend indicating an increase in growing season average temperature of about 0.6 °C and lower precipitation over the recorded 45 years, while for Saskatchewan, the long-term trend suggests lower growing season temperatures and higher precipitation. The differences in climate patterns across the Canadian provinces can be explained by different topographical areas, with mountains casting a fairly substantial rain shadow (e.g., western border of Alberta) and open plains allowing cold arctic air to rush through the provinces (e.g., Saskatchewan and Manitoba).

While global warming is frequently associated with increased average temperatures, one has to consider that increased variability in temperatures and precipitation is also the result of global warming. This increased variability translates into increases in the frequency, intensity, spatial extent, duration, and timing of weather extremes. More frequent and more intense extreme weather events, such as floods, droughts, and heat waves, may require both ongoing farmers' response through the implementation of relevant adaptation strategies and policy makers' response through the implementation of policies that would facilitate the adoption of such strategies. A better understanding of the historical impact of weather on productivity and inefficiency in agriculture is important for producers and policy makers to fully embrace the impact of changing climate and design strategies to mitigate the effects of unfavourable weather events. So, the next section sets up the theoretical framework to assess the relationship between climate and productivity/efficiency in crop production in Canada.

### 2.2. Theoretical Model Specification

The existing literature commonly follows two approaches to estimate the impact of weather shocks on agriculture. One approach relies on partial productivity analysis where crop yields are regressed on climatic variables [18–20]. The other approach is based on the Ricardian model that establishes the relationship between land values (profitability) and climate [21]. In this paper, to estimate the impact of climate change on crop production

and technical inefficiency, we adopt the stochastic frontier production model proposed independently by [22,23].

A stochastic frontier production model requires econometric estimation of a production function. For clarity of presentation, first, we would like to define the main concepts including production function and efficiency/inefficiency. Every production process requires inputs (e.g., fertilizer, land, workers, machinery) to produce outputs (e.g., corn, wheat, potatoes). A production function shows the relationship between inputs and outputs. The curve plotting the maximum quantity of output that can be produced from a given quantity of an input (or combination of inputs) is called a production frontier. Figure 1 illustrates the production frontier where it is assumed for simplicity that in order to produce output (corn), only one input (land) is required. Farm A is on the frontier, indicating that if 40 acres of land are used, the maximum amount of corn production amounts to 7 tons. Farm A would be considered best practice. Alternatively, if one looks at Farm B—a point below the frontier—then it produces less than what is potentially possible, meaning that it is not putting its resource (land) to its best possible use. Farm B would therefore be considered inefficient.

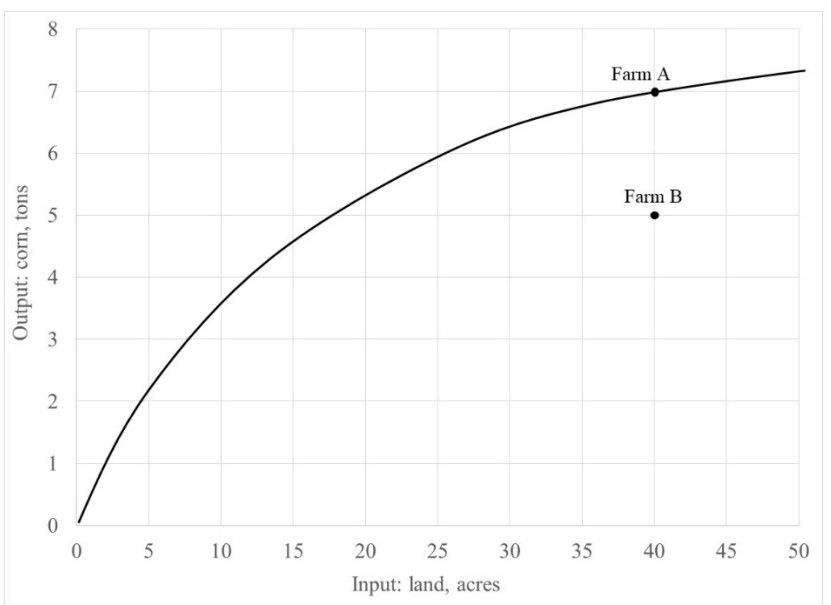

**Figure 1.** Production frontier relating output based on input.

The definition of efficiency may differ across different sciences. In engineering, for example, efficiency is defined (and measured) by the "output-to-input ratio". In applied science, generally a measure of efficiency would be obtained by comparing the observed value compared to a "reference" value, with a value of unity obtained when the two are equal. For example, in combustion engineering, efficiency is "the ratio of the amount of heat liberated in a given device to the maximum amount which could be liberated by the fuel (being used)". Within the production possibility frontier framework, if a farm is located on the frontier (such as Farm A in Figure 1) it is said to be efficient. Technical efficiency is a concept that expresses the degree to which the observed farm (or firm) performance approaches its potential. Observations (farms) below the frontier are inefficient (e.g., Farm B), with the degree of inefficiency measured by the distance between Farm A (best practice) and Farm B. For any given amount of input used, an observation on the frontier serves as a reference: for this observation, the value for efficiency equals 1 or, alternatively, the value for inefficiency equals 0. For example, from Figure 1 above, when 25 acres of land is employed in corn production, a farm that produces 6 tons of corn suffers no inefficiency, i.e., it is efficient. Technical efficiency is affected by the size of operations

and by managerial practices. In our work, in the context of crop production, technical efficiency is postulated to also be affected by weather and climate.

Econometrics methods can be used to estimate the production possibility frontier for a given dataset. The general specification panel stochastic production frontier is given as:

$$lny_{it} = lnf(x_{it}, \beta) + v_{it} - u_{it}, \tag{1}$$

where $ln\ y_{it}$ is the natural logarithm of the observed output of a cross-section unit $i$ at time $t$; $lnf(x_{it}, \beta)$ is a translog function representing the maximum quantity of output (production frontier); $x_{it}$ is a vector of the determinants of production (such as capital, labour, energy, and raw materials, all given in natural logarithms) and their interactions; $\beta$ is a vector of coefficients; $v_{it}$ is the idiosyncratic error term; and $u_{it}$ is the one-sided error term representing inefficiency. The two error components, $v_{it}$ and $u_{it}$, are independently and identically distributed as formally presented below. The translog stochastic production frontier is explicitly given as:

$$lny_{it} = \alpha_i + \sum_j^J \beta_j lnX_{jt} + \frac{1}{2}\sum_j^J \beta_{jj}(lnX_{jt})^2 + \sum_j^J\sum_k^K \beta_{jk}lnX_{jt}lnX_{kt} + v_{it} - u_{it}, \tag{2}$$

The production is technically efficient if it occurs on the production frontier and the deviation from the frontier is only due to a random statistical error given by $v_{it}$. However, a cross-sectional unit (e.g., a firm or a farm) may not be able to achieve the maximum production quantity and, therefore, deviate from the production frontier due to productive inefficiency represented by $u_{it}$ such that $u_{it} \geq 0$, which is treated as a one-sided error term in Equation (1). It is assumed that the two error terms are independent of each other.

Equation (1) involves two separate error terms, unlike a standard econometric specification with a single error term, and, as a result, requires a different estimation approach. It is assumed that the standard *i.i.d.* (independently and identically distributed, with zero mean and variance $\sigma_v^2$) assumption applies to the stochastic error term, $\left(v_{it} \sim_{iid} N\left(0,\sigma_v^2\right)\right)$. Under this assumption, Equation (1) would be amenable for estimation using standard panel fixed or random effect methods if there was no additional one-sided error term. However, with the presence of the inefficiency-related error term, one has to assume a distribution for $u_{it}$. There are various distributional assumptions underlying the one-sided error term $u_{it}$; namely, exponential, half normal, truncated normal, and gamma. The half-normal distribution assumes the distribution is truncated at mean zero; that is, $u_{it} \sim_{iid} N^+\left(0,\sigma_u^2\right)$. This distributional assumption is not used in studies that are primarily interested in modelling the determinants of inefficiency. The truncated normal is relatively general given that the truncation is not necessarily set at zero but at the mean given by $\mu$, and is given as $u_{it} \sim_{iid} N^+\left(\mu,\sigma_u^2\right)$. This allows modelling the mean of inefficiency as a function of explanatory variables, making it the ideal assumption in studies such as ours. Because the probability density function, which is a function of the mean and variance of the inefficiency, is used to formulate the log-likelihood function, estimation results could differ based on the underlying distributional assumption. The gamma distribution is given as:

$$h(u_i) \sim \left(\frac{\theta^p}{\Gamma(p)}\right)u_i{}^{p-1}exp\{-\theta u_i\},\ u_i \geq 0, \sigma_u^2 = \frac{1}{\theta}, \text{ and } p > 2. \tag{3}$$

and it is simply the exponential distribution when $p = 1$, leading to $h(u_i) \sim \theta exp\{-\theta u_i\}$ [24]. Obtaining estimates for $u_{it}$ is based on the conditional mean ($E[u_{it} | \epsilon_{it}]$), where $\epsilon_{it} = v_{it} - u_{it}$. This conditional expectation function takes different forms for different distributional assumptions discussed above. For example, for the normal/half-normal distribution $\left(v_{it} \sim_{iid} N\left(0,\sigma_v^2\right) \text{ and } u_{it} \sim_{iid} N^+\left(0,\sigma_u^2\right)\right)$, the conditional expectation is given as $E[u_{it}|\epsilon_{it}] = \frac{\sigma\gamma}{1+\gamma^2}\left(\frac{\varphi(z_i)}{1-\Phi(z_i)} - z_{it}\right)$, where $z_{it} = \frac{\epsilon_{it}\gamma}{\sigma}$; $\gamma = \frac{\sigma_u}{\sigma_v}$; $\sigma = \left(\sigma^2\right)^{\frac{1}{2}}$, and $\sigma^2 = var(\epsilon_{it}) = var(u_{it}) + var(v_{it}) = \left(1 - \frac{2}{\pi}\right)\sigma_u^2$. In [25], the authors use this result

to define technical efficiency as $TE_{it} = e^{-E(u_{it}|\epsilon_{it})}$, making technical inefficiency simply $TI_{it} = 1 - e^{-E(-u_{it}|\epsilon_{it})}$.

The two important issues related to this study are (i) our model is panel, which requires additional considerations regarding the behaviour of inefficiency across time, and (ii) this paper also aims to model the impact of climate on farm technical inefficiency, and this task requires consideration of the techniques used to model inefficiency as a function of exogenous factors. Several different specifications have been recommended for estimates of technical inefficiency using the panel data stochastic frontier approach. Early in the development of this approach, [26] proposed an estimation of technical inefficiency based on a fixed effects model in which the inefficiency is time-invariant and is estimated by the individual specific constant term. This suggests inefficiency can be measured as the deviation of the *i*th producer's specific intercept from the maximum within the sample, with $\hat{\alpha}_i^* = \max(\hat{\alpha}_i) - \hat{\alpha}_i$. This approach has two key limitations. First, it attributes all individual heterogeneity to inefficiency. Second, it assumes that inefficiency is static and does not evolve over time. The authors of [27–30] all propose alternative stochastic frontier panel specifications that allow for the evolution of inefficiency over time.

In all these specifications, all panels share the same intercept term, $\alpha_0$, which implies that time invariant individual effects are included in the inefficiency estimate. Attributing all individual specific effects to inefficiency will likely result in biased inefficiency estimates, so the common issue is implicit attribution of all unexplained heterogeneities to differences in inefficiencies. In [31,32], the author proposes two different approaches: namely the "true" fixed effect and "true" random effects (TFE and TRE, respectively) specifications that disentangle the time-varying inefficiency from the time-invariant individual heterogeneity effects. Stata$^{®}$ commands for the estimation of TRE and TFE are offered by [33].

The second consideration vital for this paper is modelling inefficiency as a function of other explanatory variables. When technical efficiency is assumed to be influenced by some exogenous factors, there exist two approaches to estimating the full model. One approach involves two steps where the stochastic frontier production function is estimated first to obtain efficiency estimates. These estimates are then used to explain the variation in efficiency in a second step. A number of papers, however, argue that the two-step procedure yields biased and inefficient estimates [34–36], with alternative approaches having been proposed where both the production and efficiency equations are estimated simultaneously in one step.

This is implemented by specifying the mean of truncated normal inefficiency as a function of the underlying factors, given as:

$$u_{it} \sim N^+\left(\mu_{it}, \sigma_u^2\right) \text{ where } \mu_{it} = h(z_{it}, \delta) \tag{4}$$

Equation (4) assumes that only the mean is dependent on exogenous variables. However, the variance of the inefficiency term can also be heteroscedastic and, therefore, dependent on other exogenous variables. The heteroscedastic version of Equation (4) is given as:

$$u_{it} \sim N^+\left(\mu h(z_{it}\delta), \sigma_u^2 h(z_{it}, \delta)\right) \tag{5}$$

The set of variables in the two functions do not have to necessarily be identical [33]. In addition to the one-sided error (the inefficiency) term, the two-sided error term could also be treated as heteroscedastic. We explore both specifications in our empirical section, where $z_{it}$ is a vector of primarily climatic variables and time trends.

## 2.3. Incorporating Climate/Weather in the General Stochastic Frontier Framework

The stochastic frontier function given by Equation (1), which commonly includes only inputs as output determinants and individual producer characteristics as inefficiency determinants, can easily be expanded to include climatic variables. The contribution of climate to crop productivity is two-fold and, therefore, climate variables are incorporated

both in the vector of inputs $x_i$ and as determinants of technical inefficiency. On the one hand, rainfall, temperature, and sun radiation can be viewed as a natural "non-cost" input, on par with purchased inputs such as seed and fertilizers because plants need both proper temperatures and rainfall to grow [37,38].

It should be mentioned that "weather" is a much more complex term than just temperature and precipitation that most studies include when assessing the impact of climate on crop productivity. However, due to the unavailability of data on humidity, sunshine, wind velocity, and other climate characteristics, temperature and precipitation are often used as the main descriptors of weather. The use of temperature and precipitation as the only explanatory variables in studies exploring the relationship between crop productivity and climate change can be justified by the fact that temperature is a primary factor affecting the rate of plant development; temperature is important during the germination stage and is very important during the pollination stage [39]. Precipitation plays an important role because temperature effects may be mitigated or aggravated by water deficits and/or excess soil water. Soil moisture is also an important factor that determines the ability of plants to extract nutrients.

Although a lot of research efforts have been dedicated to study the relationship between weather factors and crop yields, this relationship is still not fully understood [38]. Some studies included temperature and precipitation in a linear fashion, while other studies have identified the non-linear relationship between weather and yields [20,21,40,41]. Most commonly, the non-linear relationship is represented by the inclusion of quadratic terms for temperature and precipitation; however, some research works have developed composite "aridity indexes" to capture the interaction between temperature and rainfall in their impact on yields [37]. In this study, we employ the aridity index to capture the importance of the interaction between temperature and precipitation and some measures of weather variability, which is discussed in greater detail below.

While climatic conditions can be thought of as a direct input into crop production and therefore should be included as inputs in the production function specification, they can also have an indirect impact on crop productivity through farmers' responses to adapt to climate change. For example, producers can try to mitigate losses by changing their input mix and/or by changing their crop mix. These producers' responses are likely to be reflected in the technical inefficiency of farms; operators in regions with more variable, harder-to-predict climatic conditions may operate relatively further from the production frontier compared to producers operating under more predictable conditions. In addition, weather conditions may influence the productivity of other inputs. This would especially be true for fertilizers as sufficient moisture is required for plants to be able to extract nutrients from soil. So, a lack of rain in a certain production region can make fertilizers much less efficient in this region compared to regions that had enough rain. This would translate into a below frontier production point (i.e., technical inefficiency) for this region. Given the potential impact of weather on technical inefficiency, climatic variables should also be included as factors explaining inefficiency.

### 2.4. Functional Form for the Production Function

In Equation (2), we proposed the more flexible translog functional form that can be used to derive quantitative estimates of the direct and indirect effect of climate variables on crop production. The choice of a functional form for the estimated frontier is an important issue arising in the empirical estimation of stochastic production frontier models. Empirical applications for the measurement of efficiency have traditionally employed a Cobb–Douglas specification or a more flexible translog function. A number of studies have evaluated the performance of different functional specifications in the estimation of technical efficiency and assessed their ability to approximate alternative technologies [42–44].

The authors of [44] state that locally flexible functional specifications, including translog, generalized Leontief, normalized quadratic, and squared-root quadratic forms, provide equally plausible *a priori* approximations of a true but unknown production tech-

nology and there is no *a priori* reason to favour any one of them. While the authors find that the choice of the functional form matters for conclusions concerning the production possibilities and the efficiency estimates, their comparative analysis also reveals that the empirical *ex post* evaluation does not always lead to the determination of the best functional specification as the search for the appropriate specification can yield statistically indistinguishable results.

In this paper, various functional forms have been estimated and the Cobb–Douglas function form, given in Equation (6) below, was the final selection for several reasons.

$$lny_{it} = \alpha_i + \sum_j^J \beta_j lnX_{jt} + v_{it} - u_{it},$$ (6)

First, while this functional form is somewhat restrictive, it is easily estimable and does not require a loss of many degrees of freedom due to the inclusion of quadratic terms and cross-products. Second, the estimated production function was well behaved at the point of approximation in a sense that it satisfied all the regularity assumptions, namely, positivity, monotonicity, and curvature. Positivity requires that the estimated output be positive for all data observations. Monotonicity requires that the first-order derivatives of the production function (i.e., marginal products) be positive. Curvature requires that the production function be quasi-concave or, equivalently, that the Hessian matrix be negative semi-definite.

One might argue that, under the assumption of profit maximization or cost minimization where producers choose inputs in response to production and market conditions, inputs are not independent of the error term and, therefore, direct estimation of the production function leads to biased and inconsistent estimates due to an endogeneity problem. The authors of [45] postulate that when stochastic elements such as weather, price uncertainties, and other unanticipated disturbances are introduced into the model, then the concept of "profit maximization", which is unambiguous within the deterministic framework of the model, is no longer applicable. Instead, a plausible assumption is that producers maximize expected profits, in which case inputs are independent of the disturbances arising due to "acts of nature" and "human error" [45]. The assumption of expected profit maximization would especially be true in agriculture where the production process is not instantaneous and the effect of the disturbance on output cannot be known until after the preselected quantities of inputs have been employed in production.

### 2.5. Data Description

The data used in this study were extracted from Statistics Canada (inputs–output in crop production) and the Meteorological Service of Canada (climate data). Our analysis is performed at a provincial level for the period 1972–2016. Due to a lot of missing information, the province of Newfoundland and Labrador is excluded from the study, thus leaving nine provinces in the analysis.

The dependent variable in Equation (6) is the annual value of cash crop receipts measured in Canadian dollars and deflated by the farm product price index. It should be noted that cash receipts may differ from the value of farm output due to the role of inventory adjustments, seed held back, and crops used on farm for feed. However, due to data availability, cash receipts are used as a proxy for farm output.

Another issue pertaining to crop output is a substantial difference in cropping patterns and types of farms in different provinces of Canada. In particular, some provinces have climate and soil conditions more suitable for the production of field vegetables including sugar beets and potatoes, while some provinces' crop output consists primarily of grains. In addition, crops produced on crop-producing farms and crops produced on livestock farms may have different productivity [46]. Limited by the availability of data, following [46], one plausible solution is to include a percentage of livestock products ($S_L$) in total output and

a percentage of output from grains and legumes ($S_{G\&L}$) in total crop output as additional explanatory variables.

To estimate the production function specified in Equation (6), one has to construct input variables. Adopting the input classification of [12], the inputs are categorized into four categories: capital (K), including machinery and equipment, depreciation on machinery, and machinery repairs, all measured in Canadian dollars; total labour (L), comprising paid labour (hired and family), measured in Canadian dollars, and unpaid labour, measured in hours per year; land (A), represented by the value of land and buildings, building repairs, building depreciation, and property tax; and intermediate inputs (M), consisting of commercial seed, twine and containers, telephone expenses, crop and hail insurance, business insurance, custom work, and other miscellaneous expenses, measured in Canadian dollars. As mentioned by [12], these four input categories provide sufficient disaggregation for econometrics analysis and at the same time are not too narrow to make the analysis problematic.

One of the difficulties faced in this study is the lack of appropriate quantity data for the production input variables. Using value rather than quantity data poses a number of challenges. First, using values produces bias due to relative input price differences that can be substantial, especially for land and labour as they will reflect differences in quality and location [46]. Using "acres" for land input as an alternative to the value of land and buildings might appear to be a plausible solution; however, this alternative unit of measurement ignores differences in the quality of agricultural land across Canadian provinces, as well as across different locations within a province. The second challenge concerning the use of values rather than quantities surrounds conversion from nominal dollars into real. Although Statistics Canada provides a Farm Input Price Index (FIPI) at the level of specific input categories, these data are given for different time periods using different base years in the construction of the index. Data inquiries with Statistics Canada revealed that there were significant changes to the methodology for FIPI throughout the years and given these changes, it was impossible to generate a linked series of reasonable quality for FIPI at the level of specific input categories spanning the study period 1972–2016. Upon our personal request, Statistics Canada was able to generate a continuous historical series only for total aggregate of Farm Inputs for Crop Production at the national level. The reference period for the historical index is 1986 = 100. One caveat of this historical index is that it does not factor into account basket changes that could have occurred over the historical period; in addition, crop insurance was not introduced to the index until 1981. So, input variables are constructed keeping these data limitations in mind, with a detailed explanation of the data provided in Appendix A.

Another set of variables important to this study includes climatic variables. In the existing literature, there is no single approach as to how to define and measure climate/weather. Some studies distinguish between "weather" and "climate" [15,47]. Climate is assessed as mean precipitation and temperature in the long term, usually a 30-year period or longer. Weather can be assessed from the viewpoint of current weather conditions such as air temperature and precipitation or from a viewpoint of shocks and anomalies in the weather patterns. The most commonly used measures of weather shocks include average maximum and minimum temperatures [18,19,48,49], intra-annual (within-year) standard deviations calculated from average daily temperature and precipitation measures [50], and (standardized) deviations of temperature and precipitation from long-term (usually 30-year) averages (normals) [47]. The authors of [47] postulate that the inclusion of both weather and climate variables is important because there is likely to be a difference in their effects since producers can adjust input decisions in response to climate but not in response to weather.

For climatic variables, we utilize commonly used measures such as average daily temperatures and total precipitation and their standard deviations to capture intra-annual (within a specific year) weather variability, as well as some additional measures such as the number of degree days [51]. In constructing climate variables, previous studies used different units of time, such as specific months, phonological periods, and growing seasons.

In this study, we use the climatic conditions during the growing season, following [18,19] who emphasize that the average growing season climate is best to capture the net effect of the entire range of the crop development process.

Some studies have shown that there are concomitant interactions between weather variables, with the impact of precipitation on plant growth being different depending on the air temperature. To allow for the interaction between temperature and precipitation, a weather index (Oury index) is constructed as a measure of climatic conditions discussed above.

The Oury index is viewed as rainfall normalized with respect to temperature and implies that crop yield response to rainfall is not constant but rather depends on the temperature. The index is calculated as:

$$W_m = \frac{P_m}{1.07^{T_m}} \tag{7}$$

where $W$ represents the weather index (Oury index); $m$ is the month ($m = 1, 2, \ldots . 12$); $P_m$ is the total precipitation for month m in millimeters; and $T_m$ is the average temperature for month $m$ in degrees Celsius. Since the focus of our study is crop production, only the primary growing season months (May–August) were considered in the construction of the weather index. For each year $t$, we computed the mean weather index for the growing season in that year, $W_t$, by first calculating the weather index for each month from May to August and then averaging the indices over the four months.

To measure the impacts of unexpected weather shocks or potential weather extremes on crop production, a short-term Oury shock index (OSI) was constructed as:

$$OSI_{p,t} = \frac{(W_{p,t} - W_{p,LR})}{\sigma_{W_{p,LR}}} \tag{8}$$

where $W_{p,t}$ is the Oury mean for year $t$ for the growing season months (May–August) for province $p$; $W_{p,LR}$ is the long-run Oury mean calculated for a 30-year period between 1971 and 2001; and $\sigma_{W_{p,LR}}$ is the standard deviation of historical (1971–2001) Oury means.

The details concerning the construction of climate variables are provided in Appendix A. The tables summarizing the variables used in the study are also presented in Appendix A.

## 3. Results and Discussion

### 3.1. Final Model Specification

As mentioned above, our model requires the estimation of two equations: one for the "stochastic frontier" component and the other one for the "inefficiency" component. A number of different models were estimated and the final specification was chosen based on theoretical validity, significance of the variables, and the model's fit to the data. As mentioned above, the Cobb–Douglas functional form was chosen.

The explanatory variables considered for the stochastic frontier component include production inputs, time trend, and climatic variables. The explanatory variables in the inefficiency effects model include area under irrigation, a percentage of output from grains and legumes in total crop output, average farm size, and climatic variables. Area under irrigation can have a multifaceted impact on inefficiency. On the one hand, area under irrigation can help mitigate the impact of insufficient precipitation, thus allowing farmers to operate closer to the frontier or, in other words, to be more technically efficient. On the other hand, there can be large inefficiencies related to water use and management in irrigated agriculture. A percentage of output from grains and legumes is included to reflect the idea that one would expect farms specializing in crop production to be more efficient in crop production than farms with significant livestock production.

It should be mentioned that while a contemporaneous measure of climatic conditions (current temperature and current precipitation) may be the most appropriate to include in the vector of inputs, this measure may not be the most suitable for analyzing the impact

of climate change on technical efficiency. Producers make both long-run (e.g., investment in irrigation systems) and short-run (e.g., the amount of fertilizer to be applied) decisions. Short-run decisions are based on the observed current climatic conditions as well as short-run weather shocks, i.e., intra-annual weather variability. Long-term decisions are usually based on the producer's assessment of long-term weather shocks (inter-annual variability), which can be captured by deviations from normal. So, to identify the effect of climate on technical efficiency, the inefficiency equation includes both short-run weather shocks, measured by the standard deviations of temperature and precipitation during the growing season, and long-run weather shocks measured by the Oury shock index (OSI) that is described in greater detail in Appendix A.

### 3.2. Estimation Results and Discussion

The estimation results show that the null hypothesis of no inefficiency effects ($H_0: \sigma_u^2 = 0$) can be rejected at the 1% significance level. Therefore, the stochastic frontier should not be reduced to an OLS model with normal errors. The estimation output for both the frontier and inefficiency regressions is reported in Table 1. We show the results of three different models, all based on the Cobb–Douglas functional form, where some control variables are varied to check the robustness of the estimated climatic impacts on productivity and inefficiency. For the production function equation, Model 1 and Model 3 control for an average farm size in a province, while Model 2 does not. For the inefficiency equations, the differences between the three models lie in the inclusion of different weather variables. More specifically, Model 3 includes the Oury index that captures climate change, while Model 1 and Model 2 include an Oury shock index (OSI) which measures the impact of unexpected weather shocks or potential weather extremes on crop production. It is hypothesized that an unexpected climate shock rather than climate change explains variations in technical inefficiency. It is assumed that when farmers observe changes in climate, reflected in the Oury index, they form their expectations about upcoming weather and make appropriate adjustments to their production processes so they can extract as much output as possible from production resources; it is the unexpected weather changes that result in the inefficient inputs use.

**Table 1.** Estimation results: production possibility frontier model using true fixed effects (truncated normal distribution).

| | Model (1) | Model (2) | Model (3) |
|---|---|---|---|
| **Frontier Equation** | | | |
| Ln (Capital) [a] | −0.036 | −0.011 | −0.053 |
| | (0.098) | (0.098) | (0.103) |
| Ln (Land) | 0.387 *** | 0.376 *** | 0.390 *** |
| | (0.040) | (0.040) | (0.041) |
| Ln (Paid Labour) | 0.227 *** | 0.247 *** | 0.221 *** |
| | (0.057) | (0.053) | (0.058) |
| Ln (Unpaid Labour) | 0.113 * | 0.058 | 0.123 * |
| | (0.065) | (0.058) | (0.066) |
| Ln (Materials) | −0.028 | 0.005 * | −0.014 |
| | (0.073) | (0.070) | (0.073) |
| Ln (Standard Deviation of Precipitation) [b] | −0.133 *** | −0.146 *** | −0.118 ** |
| | (0.051) | (0.053) | (0.053) |
| Ln (Oury Index) | 0.111 ** | 0.119 *** | 0.081 * |
| | (0.043) | (0.045) | (0.045) |
| Ln (Farm Size) | 0.187 * | - | 0.207 ** |
| | (0.103) | | (0.105) |
| Trend dummy: 1981–1989 | −0.037 | −0.038 | −0.039 |
| | (0.034) | (0.034) | (0.035) |
| Trend dummy: 1990–1998 | −0.157 *** | −0.151 *** | −0.162 ** |
| | (0.048) | (0.048) | (0.048) |

**Table 1.** *Cont.*

|  | Model (1) | Model (2) | Model (3) |
|---|---|---|---|
| Trend dummy: 1999–2007 | −0.131 ** | −0.125 ** | −0.136 ** |
|  | (0.054) | (0.054) | (0.055) |
| Trend dummy: 2008–2016 | −0.072 | −0.048 | −0.080 |
|  | (0.056) | (0.056) | (0.057) |
| **Inefficiency equation** ($\mu$) |  |  |  |
| OSI | 0.042 ** | 0.043 ** | - |
|  | (0.017) | (0.017) |  |
| Ln (Oury Index) | - | - | 0.080 |
|  |  |  | (0.084) |
| Ln (Standard Deviation of Temperature) | 0.191 * | 0.236 ** | 0.181 * |
|  | (0.109) | (0.104) | (0.110) |
| Ln (Standard Deviation of Precipitation) | −0.256 *** | −0.249 *** | −0.183 * |
|  | (0.095) | (0.095) | (0.100) |
| Time trend | −0.031 *** | −0.030 *** | −0.031 *** |
|  | (0.003) | (0.003) | (0.004) |
| Percentage of irrigated area | 0.006 | 0.005 | 0.007 * |
|  | (0.004) | (0.004) | (0.004) |
| Percentage of output from grains/legumes in total crop output | 0.006 *** | 0.006 *** | 0.007 *** |
|  | (0.001) | (0.001) | (0.001) |
| Ln (Degree Days) | 0.074 * | 0.058 | 0.018 |
|  | (0.042) | (0.040) | (0.051) |
| $\sigma_u^2$ |  |  |  |
| Ln (Standard Deviation of Temperature) | 2.837 ** | 3.348 ** | 2.645 * |
|  | (1.478) | (1.653) | (1.443) |
| Ln (Standard Deviation of Precipitation) | −1.112 | −2.309 *** | −0.908 |
|  | (0.901) | (0.868) | (0.905) |
| Ln (Farm Size) | 0.667 ** | - | 0.740 ** |
|  | (0.294) |  | (0.296) |
| Constant | −11.609 *** | −6.696 *** | −11.999 *** |
|  | (3.278) | (2.53) | (3.249) |
| $\sigma_v^2$ |  |  |  |
| Constant | −5.007 *** | −4.925 *** | −5.011 *** |
|  | (0.125) | (0.117) | (0.133) |
| Observations | 405 | 405 | 405 |
| Wald chi2 (12) | 857.81 | 878.15 | 861.87 |

Note: *** $p < 0.01$, ** $p < 0.05$, * $p < 0.10$. Robust standard errors in parentheses. [a] Ln is the natural logarithm. [b] Standard deviations for temperature and precipitation are calculated from daily weather data during the May–August growing season.

Given the Cobb–Douglas functional form of the production function, the coefficients reported in Table 1 are production elasticities for individual input categories. The estimated coefficients have the expected signs except the coefficient on capital and the coefficient on materials in Model 1 and Model 3. These coefficients, however, are statistically insignificant. The coefficient on unpaid labour in Model 2 is also found to be statistically insignificant. One plausible explanation for some of the major inputs such as capital and unpaid (operators and unpaid family) labour appearing insignificant is measurement errors in the data. As is mentioned in Appendix A, the major weakness of the available dataset is that input use is not segregated between crop and livestock production. Using various input attribution methodologies may have resulted in measurement errors that mask the importance of inputs in explaining the variation in output. The estimated output elasticity of land is the highest (0.36–0.39), followed by output elasticity of paid labour (0.22–0.25). In Model 1 and Model 3, where farm size is included as a control variable, average farm size in a province is positively related to crop output, indicating that provinces with larger farms are able to extract more output from a given set of resources.

Instead of using a simple time trend that assumes a constant rate of technological change across the entire sample period, in the frontier equation we allow the rate of technological

change to vary every 8–9 years. The results indicate that the rate of technological change stayed about the same during the 1972–1989 and 2008–2016 periods; however, it declined in 1990–1998 and 1999–2007 compared to the rate experienced during the 1972–1980 period.

The key research objective of this paper is to establish the impact of weather on maximum output (production frontier) and technical inefficiency. The parameter estimates show that climatic variables are important determinants in both the frontier equation (production function) and inefficiency equation. The results reveal that the combined effects of higher temperature and lower precipitation that result in a lower Oury index drive the maximum possible crop output down. More specifically, a 1% increase in the Oury index leads to a statistically significant 0.11% reduction in output. Similarly, a 1% increase in intra-annual standard deviation of precipitation during the growing season, indicating that precipitation is less evenly distributed across the days of the growing season, leads to a statistically significant 0.13–0.15% decline in output. The results are consistent across the three models. The intra-annual standard deviation of growing season temperatures was not found to be a significant determinant of output and, as a result, was excluded from the frontier equation in the final specification.

The estimation results for the inefficiency equation indicate that both inter-annual and intra-annual variations in temperature and precipitation are important in explaining why provincial crop production deviates away from its best performance. According to the estimates, greater intra-annual variations in temperature increase inefficiency, while greater intra-annual variations in precipitation reduce inefficiency. One possible explanation why greater variations in temperature and precipitation have the opposite effects is that temperature cannot be controlled by producers. For example, if there is an unexpected heat wave, there is not much that producers can do to protect their crop from heat. The same applies for cold spells. As a result, high fluctuations in daily temperatures during the growing season lead to less efficient output. With precipitation, however, the story is slightly different as the amount of water can be controlled through irrigation and/or other production practices (e.g., no-till to keep the soil covered with crop residue, thus retaining more moisture). If farmers expect dramatic variation in precipitation throughout the growing season in advance based on their past experience, they may have already invested in an irrigation system to dampen the impact of climate change on farm production. This could explain why provinces with a higher standard deviation of precipitation are found to operate closer to the production frontier.

The results in Table 1 also reveal that the deviation from the historical normals (OSI) has more pronounced impact on production efficiency than the mean level changes of weather variables, measured by the Oury index. This finding is consistent with our expectation that it is unexpected weather events that set producers further away from the production possibility frontier rather than climate change that is anticipated by producers.

As for the impact of other control variables on production inefficiency, we find that the percentage of output from grains and legumes in total crop output increases inefficiency, indicating that provinces with more focus on grains/legumes operate further away from the production possibility frontier. One explanation could be that farms that focus on fruit and vegetable production have better adapted to changing weather by investing in irrigation systems or greenhouses, for example. As a result, they are better able to apply their resources to the best possible use.

As is evident from the estimated coefficient on time trend in the inefficiency equation, inefficiency has been declining over time. This can also be seen from Figure 2 and Figure 4 below; all three models have given us very similar results. Therefore, for the sake of presentation clarity, the graphs show the results from Model 1 and Model 2 only. The average technical efficiency score in the 1970s was in the 0.55–0.7 range, while in recent years the average technical efficiency score has been in the 90-range, indicating that the inefficiency gap across the provinces has narrowed. Figure 3 illustrates the mean technical efficiency scores by provinces over time. The results suggest that the Prairie provinces have been the slowest to catch up to the production possibility frontier.

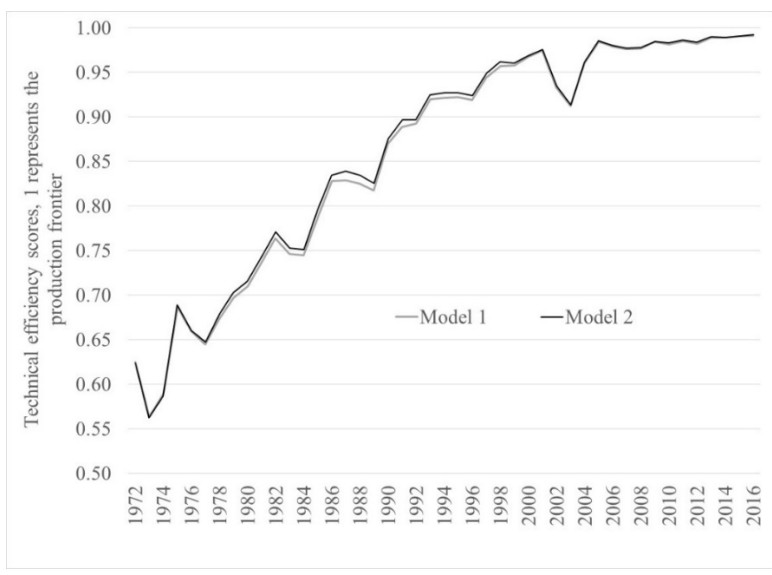

**Figure 2.** Average technical efficiency scores over time.

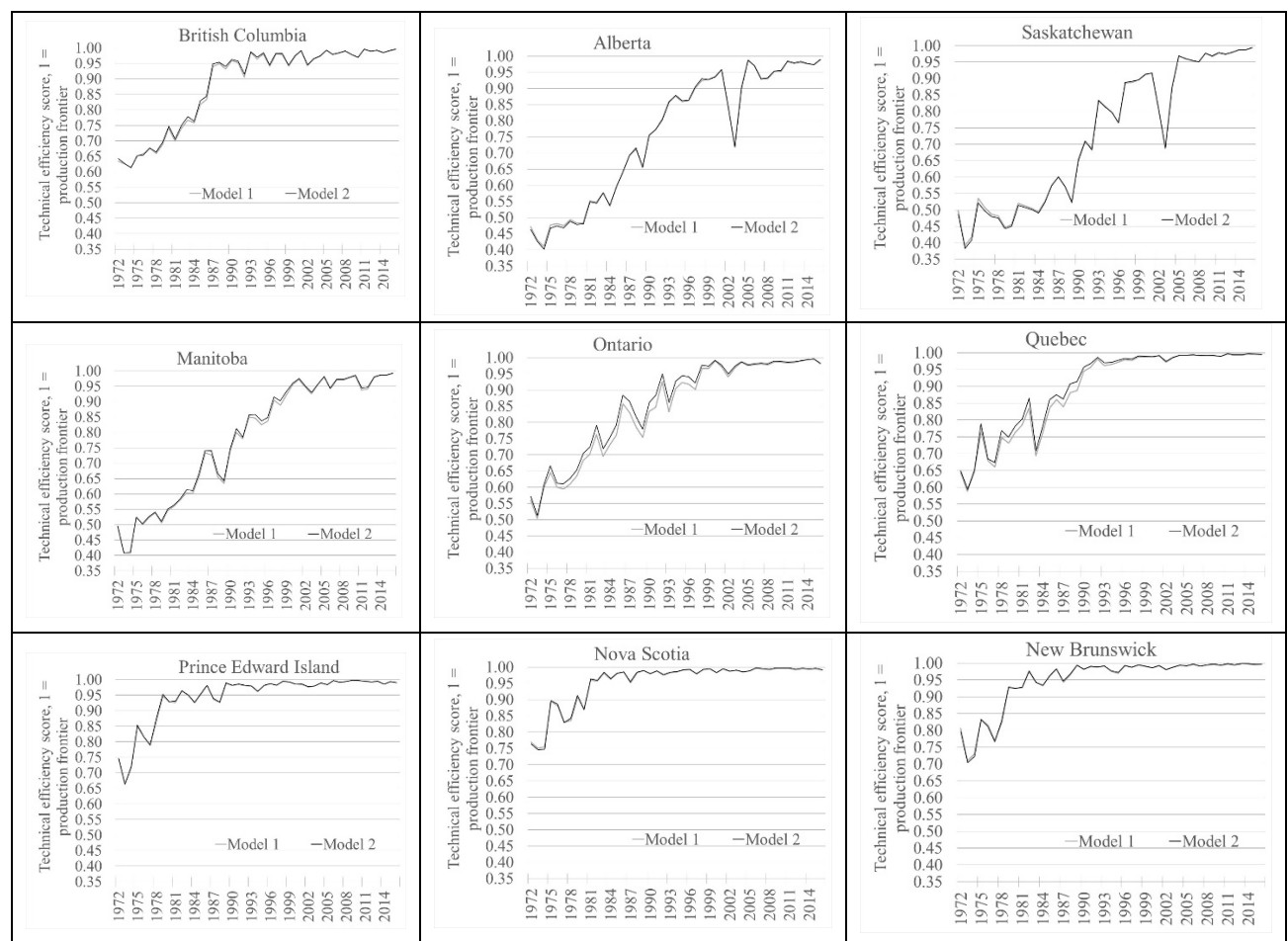

**Figure 3.** Provincial trends in technical efficiency scores.

The rankings of the provinces in terms of mean efficiency scores throughout the entire study period are presented in Figure 4. We find that over the study period, the Atlantic provinces make it to the list of the most efficient provinces, while the Prairie provinces (SK, MB, and AB) operate relatively further away from the production frontier. This ranking

is consistent with the discussion above regarding how the share of crop production is positively related to inefficiency; more specifically, the production of fruit and vegetables is more efficient.

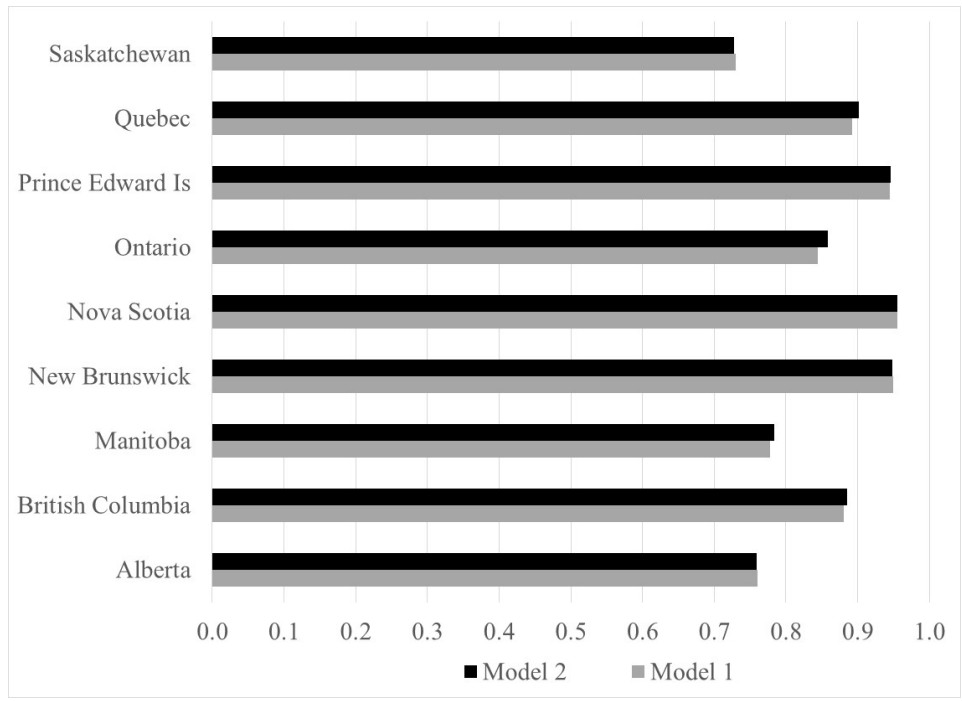

**Figure 4.** 1972–2016 average technical efficiency scores by province.

## 4. Conclusions and Policy Implications

This study explores the effects of climate change on agricultural productivity and technical efficiency across Canadian provinces. To the best of the authors' knowledge, it is the first such study in the context of Canada. To capture climatic conditions faced by crop producers in Canada throughout 1972–2016, this study constructed a unique dataset using weather information at a level of individual weather stations.

The production frontier and technical inefficiency equations were estimated using panel stochastic frontier specification with heterogeneous means and variances of the technical inefficiency. Our approach was based on the true fixed effects model which disentangles the provincial fixed effects from the time-varying technical inefficiency. To take into consideration the fact that plant growing conditions such as temperature and precipitation serve as direct inputs into the production process alongside other major inputs such as capital, land, and labour, and at the same time can potentially influence productivity of other inputs, thus explaining why some producers are more efficient than others, weather variables are included in both the frontier and technical inefficiency equations.

Having carried out several specifications and combinations of the variables, we arrived at three different models meeting the conditions for theoretical validity. Our approach from general to specific search for the best fit generated a robust and consistent prediction showing that weather and climate are important predictors of both maximum possible output and technical inefficiency. More specifically, our results have shown that higher temperatures coupled with reduced precipitation create less favourable conditions for crop production in Canada. The results also reveal that reducing the weather shocks could significantly benefit crop production.

Based on our findings, a number of policy implications arise. First, given that climate change is expected to result in increased frequency of unfavourable events and more weather variability in the future, investment in public research will be very important to mitigate the negative impacts of future climate change. With climate change, farmers need

science more than ever, yet public funding for research that can help them cope has been in short supply. Agroecology research, which has been particularly underfunded, has the potential to play an important role in helping farmers adapt to climate change and stay viable for generations. The impacts of changing climate on crop production are real and significant, so farmers have to be informed and educated about growing practices that can mitigate such impacts. Investing in the development of crops that are more tolerant to droughts, warmer temperatures, and more resistant to various pathogens will be very important to ensure that farmers are able to withstand climate change.

The results revealed that the intra-annual standard deviation of precipitation has a significant adverse effect on maximum crop output. This finding suggests that investment in infrastructure such as irrigation could mitigate the negative impacts of climate change by ensuring that precipitation is more evenly distributed across the growing season rather than falling in a few episodes, or not at all. While temperature is completely outside farmers' control and would have to be, therefore, taken as given, the amount of water available to plants can be controlled via the introduction of irrigation techniques. So, if irrigation infrastructure is in place, the unfavourable effects of warmer temperatures combined with dry conditions could be offset by providing more irrigation water to crops, thus maintaining a suitable precipitation/temperature (Oury) index for optimum plant growth.

**Author Contributions:** Conceptualization, V.G. and S.G.; methodology, V.G. and S.G.; software, S.G.; econometric estimation, S.G.; formal analysis, V.G. and S.G.; resources, V.G.; data curation, V.G.; writing—original draft preparation, V.G.; writing—review and editing, V.G.; visualization, V.G. and S.G. All authors have read and agreed to the published version of the manuscript.

**Funding:** This research received no external funding.

**Institutional Review Board Statement:** Not applicable.

**Informed Consent Statement:** Not applicable.

**Data Availability Statement:** The data can be access from the authors upon request.

**Acknowledgments:** We would like to thank Statistics Canada, Liam Moore (Unit Head, Producer Prices Division) and Carter Thompson (Consulting Analyst, Statistics Canada), more specifically, for providing us with a timely response and assistance in collecting input–output data. We would also like to thank the Meteorological Service of Canada, Environment and Climate Change Eastgate Offices, for providing access to detailed station-level weather data. We would also like to thank the University of Regina for partially covering the cost of publication of this article through the President's Publication Fund.

**Conflicts of Interest:** The authors declare no conflict of interest.

## Appendix A. Detailed Data Description

*Appendix A.1. Climatic Data*

As discussed above, input–output data from 1972 required for an estimation of a production function are available only at the provincial level, while climate data are available by meteorological station rather than by province. So, it was necessary to estimate province-level climates. Within one province, weather can significantly fluctuate across different locations, for example, census agricultural regions (CARs). In addition, some locations within each province are more suitable for agricultural production than others, so taking a simple average of climates across the stations in a particular province might not reflect the environmental conditions faced by farmers. Given the availability of data on cropped land by CARs, the provincial weather data are constructed from CARs-level climates weighted by the share of crop land for those regions. Therefore, climatic conditions in census agricultural regions that contribute more to the provincial crop production are given more weight in constructing the overall province-level climate variables.

The construction of CARs-level climates began with climate data obtained from the Meteorological Service of Canada, Environment and Climate Change Canada Eastgate

Offices, which gathers data from more than 8000 weather stations across Canada. The data include information on average, minimum, and maximum temperature and precipitation for each day from 1972 through 2016 for each weather station. Station-level climates are used to estimate average temperature and precipitation for each day of the year in a specific CAR. The average daily numbers are then used to calculate the average temperature and total precipitation for each month of the year at a CAR level.

In order to link the agricultural data which are organized by provinces and the climate data which are organized by stations, geographic coordinates and mapping software were used to link weather stations to the CARs. It should be mentioned that in some provinces (QB, SK, AB), census agricultural regions changed either their names or their boundaries between census years 1986 and 2016, which made it difficult to reconcile the CARs-level data across different census years. Boundary files for 2006 census agricultural regions were readily available, so this analysis is based on 2006 CARs. Since the distribution of crop land across different CARs has changed little between 1981 and 2016, the 2006 share of crop land across the CARs is used to construct weights for climate data as explained above.

The climate variables most commonly used in the existing literature include total precipitation, average temperature, and minimum and maximum temperatures. So, in this study, the set of weather variables comprises: (1) province-level average growing season (May–August) temperature, (2) total precipitation for the crop growing season, and (3) intra-annual standard deviation of temperature and precipitation to capture shocks and anomalies in the weather patterns within a given year.

In addition to the above-mentioned weather variables, a measure of degree days has been included. The agronomic literature has found that plants respond in linear fashion to temperatures within a certain temperature range, while temperatures below or above this range become harmful for plant growth. A concept of degree days, measured as the sum of degrees above the lower threshold and below the upper threshold during the growing season, has been used to capture the relationship between plant growth and temperature. We follow [51] in calculating degree days, where 8 °C and 32 °C are taken as the lower and upper bounds.

The above set of climate variables is used to create weather indexes. Some studies have shown that there are concomitant interactions between weather variables. For example, the impact of warmer temperatures may be different depending on the amount of rain. The author of [37] proposed that in the construction of crop production models, it would be useful to use aridity indexes to capture temperature–rainfall interactions and their combined impact on plant growth. In [38], the authors provide an extensive discussion of various aridity indexes developed in the literature and find that the crop production model utilizing the Angstrom index [52] performed better than other index models. In this study we employ Angstrom's weather index, which is represented by the following equation:

$$W_m = \frac{P_m}{1.07^{T_m}} \tag{A1}$$

where $W$ represents the weather index (Oury index); $m$ is the month ($m$ = 1, 2, ... 12); $P_m$ is the total precipitation for month $m$ in millimeters; and $T_m$ is the average temperature for month $m$ in degrees Celsius. The weather index can be viewed as rainfall normalized with respect to temperature. The index implies that crop yield response to rainfall is not constant but rather depends on the temperature. While livestock production is influenced by animals' year-round thermal environment, crop production is heavily affected by precipitation and temperature during the growing season. Since the focus of our study is crop production, only the primary growing season months (May–August) are considered in the construction of the weather index. For each year t, we compute the mean weather index for the growing season in that year, $W_t$, by first calculating the weather index for each month from May to August and then averaging the indices over the four months.

It is important to note that climate can be assessed from a viewpoint of long-term climatic conditions (normals) (variations in climate) or short-term climate variations (weather

shocks or variations in weather), which represent the annual climatic deviations from long-term conditions. The long-term Oury index (normal) was generated for a 30-year period spanning from 1971 to 2001. Short-term climate data are calculated as the deviation from long-term means divided by the standard deviation calculated from the 1971–2001 data. More specifically, to measure the impacts of unexpected weather shocks or potential weather extremes on crop production, a short-term Oury shock index (OSI) was constructed as:

$$OSI_{p,t} = \frac{(W_{p,t} - W_{p,LR})}{\sigma_{W_{p,LR}}} \tag{A2}$$

where $W_{p,t}$ is the Oury mean for year t for the growing season months (May–August) for province $p$; $W_{p,LR}$ is the long-run Oury mean calculated for a 30-year period between 1971 and 2001; and $\sigma_{W_{p,LR}}$ is the standard deviation of historical (1971–2001) Oury means.

*Appendix A.2. Input and Output Data*

Following [12], the inputs are classified into four broad categories: capital, labour, land, and intermediate inputs (materials). Capital input includes machinery and equipment (M&E) expenses, depreciation on M&E, and M&E repairs. Labour consists of paid and unpaid labour. While data on cash wages (hired labour and paid family labour) are available from Statistics Canada, an important part of total labour input is labour input from operators and unpaid family members. In order to estimate labour input from farm operators, an approach used by [10] is followed. More specifically, the operator's input is derived from questions on the censuses of agriculture about off-farm work and, in most recent censuses, about on-farm work. Unpaid family labour is then calculated as 70% of the operator's labour input, following [11]. Total unpaid labour is found as the sum of the main operator's hours and unpaid family labour hours. Given the changes in the type of questions asked in the census questionnaires and allowance for more than one operator starting with the 1991 census, derivations are performed separately for the 1971–1986, 1991, and 1996–2016 census years. Data for intercensal years are estimated using linear interpolation between estimates for census years.

The decomposition of the various input categories in crop production and the data sources are summarized in Table A1. Table A1 also provides the units of measurements of the variables involved. For variables measured in current dollars, implicit quantities are derived by deflating the current dollar series with the relevant deflator.

Since the focus of our study is on crop production, agricultural inputs must be subdivided between crop and livestock production. While for inputs such as feed, seed, fertilizers, etc., the allocation to crop or livestock operations is obvious, other inputs such as capital, buildings, and labour cannot easily be attributed to a specific activity. So, allocating inputs between livestock and crop activities warrants some discussion.

The author of [12] allocates inputs to crop and livestock production based on a number of measures derived from Statistics Canada's Census of Agriculture, including the share of cropped land in crop and livestock operations and the share of livestock capital in crop and livestock operations.

In [12], the share of M&E expenses on crop and livestock farms is used to allocate all of the capital inputs between the two sectors. More specifically, to find the overall sectoral share for crops which includes all crop activities, the share of M&E devoted to crop activities in the livestock sector ($ML_C$) is calculated as:

$$ML_C = MC\left[\frac{CL}{CC}\right] \tag{A3}$$

The share of M&E devoted to livestock activities in the crops sector ($MC_L$) is calculated as:

$$MC_L = ML\left[\frac{LC}{LL}\right] \tag{A4}$$

**Table A1.** Variables involved and their data sources.

| Variable | Source |
|---|---|
| **Output** | |
| Farm cash receipts from crop production (nominal value), thousand CAD | Statistics Canada, CANSIM 002-0001 (Table 32-10-0045-01) |
| Farm product price index, crop production, 2007 = 100 | Statistics Canada, CANSIM 002-0069 (Table 32-10-0099-01) |
| **Inputs** | |
| Farm input price index, crop production, 1986 = 100 | Statistics Canada, generated by Statistics Canada upon the authors' request |
| **Capital** | |
| Machinery and Equipment (M&E), thousand CAD | Statistics Canada, CANSIM 002-0007 (Table 32-10-0050-01) |
| Depreciation on M&E, thousand CAD | Statistics Canada, CANSIM 002-0005 (Table 32-10-0049-01) |
| Machinery repairs, thousand CAD | Statistics Canada, CANSIM 002-0005 (Table 32-10-0049-01) |
| **Land** | |
| Value of land and buildings (L&B), thousand CAD | Statistics Canada, CANSIM 002-0007 (Table 32-10-0050-01) |
| Depreciation on buildings, thousand CAD | Statistics Canada, CANSIM 002-0005 (Table 32-10-0049-01) |
| Repairs to buildings and fences, thousand CAD | Statistics Canada, CANSIM 002-0005 (Table 32-10-0049-01) |
| Property taxes, thousand CAD | Statistics Canada, CANSIM 002-0005 (Table 32-10-0049-01) |
| **Labour** | |
| Paid labour: cash wages (family and non-family wages) and custom work, thousand CAD | Statistics Canada, CANSIM 002-0005 (Table 32-10-0049-01) |
| Operator labour and unpaid family labour, hours per year | Derived from Census of Agriculture, 1971–2016, using Cahill and Rich's (2012) approach |
| **Materials** | |
| Electricity; fuel; fertilizer and lime; pesticide; commercial seed; telephone; twine, wire and containers; crop and hail insurance; business insurance; custom work; other operating expenses; irrigation. | Statistics Canada, CANSIM 002-0005 (Table 32-10-0049-01) |

The overall share of M&E used for crop production (*SMC*) is then computed as:

$$SMC = MC + ML_C - MC_L \tag{A5}$$

where *MC* and *ML* are the respective dollar shares of machinery and equipment in the crop and livestock sectors, respectively; *CC* and *CL* are the respective shares of cropped land acreage in the crop and livestock sectors, respectively; and *LC* and *LL* are the respective shares of livestock in the crop and livestock sectors, respectively.

The methodology proposed in [12] to allocate capital inputs to crop production, however, suffers from an implicit assumption that the cost per acre of cropped land in the crop sector is the same as that in the livestock sector, which is not supported by the collected census data. For example, the above equations suggest that if livestock operations use twice as much cropped land as crop operations (i.e., *CL/CC* = 2), then the crop-related expenses in livestock operations would be twice the expenses in crop operations. For some provinces, this assumption leads to calculated shares for crop-related activities in excess of 1. For example, for Quebec, using the 1971 Census data, the value of M&E on crop farms was $36.9 million for 0.3 million acres of cropped land and $3.7 million worth of livestock and poultry. The value of M&E on livestock farms was $306.1 million for 3.03 million acres of cropped land and $375.6 million worth of livestock and poultry. Equations (A1)–(A3) imply that the crop-related M&E expenses in the livestock sector should be $36.9 * (3.03/0.3) = $372.69, which is in excess of the actual M&E expenses in

livestock operations. Allocation of labour hours using the census shares methodology faces the same problem of getting shares that are negative or in excess of 1.

Taking into consideration this weakness of the approach suggested in [12], as well as the availability of data, inputs used in both crop and livestock activities are allocated to crop production using the share of cash receipts from crop production in total cash receipts.

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
