# Peer review of "Impact of Climate Change on Productivity and Technical Efficiency in Canadian Crop Production"

_sustainability, doi:10.3390/su14074241_

Round 1

Reviewer 1 Report

Line 25: Please make sure you follow the journal format. References need to be numbered.

Line 149: It should be “Figure 4 and Figure 5” instead of “Figures 5 and Figure 6”.

Line 813: Make sure you follow the journal format.

Author Response

The reviewer 1's comments have all been addressed. More specifically, references are now numbered in accordance to the journal format. The typo on line 149 has been fixed: as is indicated by the reviewer it should have been Figure 4 and 5.

Reviewer 2 Report

Impact of climate change on agriculture in Canada

This paper is fundamentally an economic analysis that is heavy on statistical modelling. The central concepts addressed in this research including efficiency and technical efficiency need to be clearly defined since readers with different academic background may have different understanding of the terms and concepts.  

Lines 135-140: So, can the authors briefly comment why the climatic parameters for the two provinces (AL and SK) are so different from each other.

Line 149: Is it Figures 4 and 5 or Figures 5 and 6?

Lines 226-240: I think it would be helpful if the authors provide a clear definition of the term efficiency or inefficiency. I come from an engineering background and I tend to think of the term efficiency as a ratio with no units. We compare output and input of water, for example, for a particular system. But, researchers from economic background compare crop output such as yield to the water input and this efficiency concept will have units.

Lines 445-450: It is hard to agree with the correlation that the presence of irrigation allows farmers to be technically more efficient compared to rain-fed agriculture. There can be large inefficiencies related to water use and management in irrigated agriculture.

Author Response

Please, see attachment.

Reviewer 3 Report

sustainability-1613495-peer-review-v1

  1. The theme of the MS, “Impact of Climate Change on Productivity and Technical Efficiency in Canadian Crop Production” is of interest if the approach used is treated with a worldwide perspective and in a clear way. However, the Abstract is not clear and the Introduction does not offer a worldwide review on the theme; instead, quotations are limited and most text is just the viewpoints of the authors without background references.
  2. The term efficiency is used in the title and abstract but is not defined in the text (Numerator? Denominator?) when it has varied meanings in cropping and irrigation. The same happens with inefficiency. Terms important for the analysis – e.g. productivity - should always be clearly defined
  3. The Introduction includes mentioning results – L83 - which are provided after objectives are given – L71. Also uncommon, the text is continued for 30 lines after objectives. Aspects referred here and in item above show a less good organization of the text.
  4. L108-110 and Fig. 1 refer to the intensity of crop production across census agricultural regions (CARs) with indicating higher revenue from crops. Higher revenue at province level because more areas are cropped or at farm level because yields are better? These are quite different perspectives and therefore should be clearly stated.
  5. L118-121: why the climate models are not identified and their results are not given in an appropriate form? These aspects are very important in the context of this paper, then avoiding just talk but showing which are the known predictions of CC
  6. L123: what are CARs? Each time that a symbol or an abbreviation is used its meaning has to be provided
  7. 2 and 3: How Temp or Prec where computed? Which were averaged, daily, monthly or annual? This info is requested also for the Figs that follow.
  8. 1, 4 and 5: some areas are divisions of the province, others seem equal to the province. Why? This has to be clarified. In addition, for the abbreviations of the provinces should also be given the names.
  9. 149: likely Fig. 4 and 5
  10. 158-160 is written: “The major impact of climate change is change in the frequency, intensity, spatial extent, duration, and timing of weather extremes that may be masked when one looks at growing season averages and/or totals.” Thus, why the analysis remain at the averages and totals level? This shows that the article is not innovative nor written for an international scientific audience.
  11. L219: the gamma distribution function should be presented as any other equation and be numbered.
  12. L281-283: telling that “due to unavailability of data on humidity, sunshine, wind velocity and other climate characteristics, temperature and precipitation are very often used as the main descriptors of weather” does not justify that your analysis only use T and P without showing that these 2 variables are enough. There is the need to better explain why it is assumed that these 2 variables produce results close to the use of all variables.
  13. Under section 3.3 the titled production function is not presented. It happens that a variety of production functions exist and the one used should be well presented
  14. Section 3.4 does not clarify enough the data used. The Oury index should be defined. The Appendix could probably be shortened and made easier to read and included in the text.
  15. Table 1 include a variety of Frontier Equations . Since results are in ln they are all small and similar. Without a previous presentation of eq. and of the expected values for given conditions, it results impossible to appreciate the results and compare the 3 models that are not described before and which results are very similar. The same happens with Fig. 6.
  16. Fig 7presents Provincial trends in technical efficiency scores but that technical efficiency is not defined. Trends look like everything is becoming good with TE tending to 1.0!

I assume that the MS is not written for a wide audience and likely not for an international journal that is scientifically motivated. With limitations pointed out above it is evident that the MS cannot be published as it is. I assume that rewriting the article and solving the referred problems the MS may be submitted again and then reappreciated.

Author Response

Please, see attachment.

Round 2

Reviewer 3 Report

  1. The authors used 4 footnotes to include replies to comments which is inappropriate
  2. Authors did not accept improving references what is strange
  3. Authors did consider that just focusing on Canada is OK without requiring to adopt a wider view; this is not appropriate for an international journal
  4. despite the word efficiency is on title, AA prefer it is only defined in section 3, which is inadequate
  5. The article would gain much if better focused on the economic model. The approach used is innovative and is not specific of Canada. Reducing to half or less the items 1 through 3 and bringing the economic model to a section 2 on material and methods would ease and improve the article.

Author Response

Please, see attached.

Round 3

Reviewer 3 Report

I have no time nor interest keeping discussing. I asked editors to find another reviewer

Author Response

Dear Editor,

we did not see any comments from the reviewer other than him saying that he would not be interested in serving as the reviewer of this manuscript anymore. So, please, give us guidance as to what we are expected to do and what kind of revisions we should undertake at this point.